# More replenishment than priming loss of soil organic carbon with additional carbon input

Junyi Liang[1,19], Zhenghu Zhou [2], Changfu Huo[3], Zheng Shi[1], James R. Cole[4], Lei Huang[5], Konstantinos T. Konstantinidis[6], Xiaoming Li[7], Bo Liu[8], Zhongkui Luo[9], C. Ryan Penton[10,11], Edward A.G. Schuur [12], James M. Tiedje[4], Ying-Ping Wang [13], Liyou Wu[1], Jianyang Xia[14,15], Jizhong Zhou [1,16,17] & Yiqi Luo[1,12,18]

Increases in carbon (C) inputs to soil can replenish soil organic C (SOC) through various mechanisms. However, recent studies have suggested that the increased C input can also stimulate the decomposition of old SOC via priming. Whether the loss of old SOC by priming can override C replenishment has not been rigorously examined. Here we show, through data–model synthesis, that the magnitude of replenishment is greater than that of priming, resulting in a net increase in SOC by a mean of 32% of the added new C. The magnitude of the net increase in SOC is positively correlated with the nitrogen-to-C ratio of the added substrates. Additionally, model evaluation indicates that a two-pool interactive model is a parsimonious model to represent the SOC decomposition with priming and replenishment. Our findings suggest that increasing C input to soils likely promote SOC accumulation despite the enhanced decomposition of old C via priming.

[1] Department of Microbiology and Plant Biology, University of Oklahoma, 770 Van Vleet Oval, Norman, OK 73019, USA. [2] Center for Ecological Research, Northeast Forestry University, 26 Hexing Road, 150040 Harbin, Heilongjiang, China. [3] Institute of Applied Ecology, Chinese Academy of Sciences, 72 Wenhua Road, 110016 Shenyang, Liaoning, China. [4] Department of Plant, Soil and Microbial Sciences, Center for Microbial Ecology, Michigan State University, 1066 Bogue Street, East Lansing, MI 48824, USA. [5] Key Laboratory of Stress Physiology and Ecology in Cold and Arid Regions, 320 Donggang West Road, 730000 Lanzhou, Gansu, China. [6] School of Civil and Environmental Engineering and School of Biology, Georgia Institute of Technology, 790 Atlantic Drive, Atlanta, GA 30332, USA. [7] International Joint Research Laboratory for Global Change Ecology, College of Life Sciences, Henan University, 85 Minglun Street, 475004 Kaifeng, Henan, China. [8] School of Geography and Remote Sensing, Nanjing University of Information Science and Technology, 219 Ningliu Road, 210042 Nanjing, Jiangsu, China. [9] CSIRO A&F, GPO Box 1666Canberra, ACT 2601, Australia. [10] College of Integrative Sciences and Arts, Arizona State University, 7271 E Sonoran Arroyo Mall, Mesa, AZ 85281, USA. [11] Center for Fundamental and Applied Microbiomics, Biodesign Institute, Arizona State University, 727 E. Tyler Street, Tempe, AZ 85281, USA. [12] Center for Ecosystem Science and Society and Department of Biological Sciences, Northern Arizona University, 600 S Knoles Drive, Flagstaff, AZ 86011, USA. [13] CSIRO Ocean and Atmosphere, PMB 1, Aspendale, VIC 3195, Australia. [14] Tiantong National Station of Forest Ecosystem, School of Ecological and Environmental Sciences, East China Normal University, 500 Dongchuan Road, 200241 Shanghai, China. [15] Institute of Eco-Chongming (IEC), 500 Dongchuan Road, 200062 Shanghai, China. [16] State Key Joint Laboratory of Environment Simulation and Pollution Control, School of Environment, Tsinghua University, 30 Shuangqing Road, 100084 Beijing, China. [17] Earth and Environmental Sciences, Lawrence Berkeley National Laboratory, 1 Cyclotron Road, Berkeley, CA 94720, USA. [18] Department of Earth System Science, Tsinghua University, 30 Shuangqing Road, 100084 Beijing, China. [19] Present address: Environmental Sciences Division & Climate Change Science Institute, Oak Ridge National Laboratory, 1 Bethel Valley Road, Oak Ridge, TN 37830, USA. Correspondence and requests for materials should be addressed to J.L. (email: liangj@ornl.gov) or to Y.L. (email: yiqi.luo@nau.edu)

Globally, a significant amount of organic carbon (C) is stored in soils. The stored soil organic C (SOC) plays an important role in regulating atmospheric $CO_2$ concentrations and climate change[1]. Priming can promote microbial growth and liberate stabilized soil C after new C additions[2,3], and therefore stimulate decomposition of old SOC[2–10]. It has been widely concerned that increased C input to soils due to rising atmospheric $CO_2$ concentrations may limit or reduce SOC storage due to the priming effect, leading to a positive feedback to climate change[7–9,11]. However, another important process, replenishment, has the potential to increase SOC via a variety of mechanisms[12–15]. The replenishment may counterbalance the priming effect. The net balance of the two processes determines the direction and magnitude of SOC change by increasing C inputs. Therefore, it is critical to quantify the two processes and the consequent net SOC change.

We quantified both the replenishment and priming effect of new C inputs by synthesizing model-extrapolated incubation experiments in which isotope-labeled C was used to trace the origins of emitted $CO_2$. Overall, 84 data sets were used for the synthesis. In this study, replenishment is the amount of new (added) C left in soil C pools after microbial respiration within a given period of time. The priming effect is the difference in C loss from old SOC between the substrate addition treatment and the control. The net SOC change is replenishment minus priming.

Before the synthesis, four models, including a conventional (i.e., first-order kinetic) decomposition model, an interactive two-pool model, a Michaelis–Menten model, and a reverse Michaelis–Menten model, were evaluated (Fig. 1). The four models represented the replenishment and priming effect with their respective assumptions (Fig. 1 and related equations). We selected the most parsimonious one using deviance information criterion (DIC)[16]. The selected parsimonious model was further validated, using two modes with either fixed or randomized parameters, as an assurance of model extrapolation beyond the observations. Eventually, we used the most parsimonious model to estimate priming, replenishment, and net C balance at a standardized time point (i.e., 1 year in this study).

Generally, the magnitude of replenishment was greater than that of priming, resulting in a net SOC accumulation after new C

input. The magnitude of the net SOC accumulation was positively correlated with the nitrogen-to-C (N:C) ratio of the added substrates. These findings suggest that increasing plant productivity and the consequent increase in C input to soils likely promote SOC accumulation despite the enhanced decomposition of old C via priming.

## Results

**Synthesis of replenishment and priming.** Our analyses showed that new C input induced priming, which on average stimulated C loss from the old SOC equivalent to 9.4% of the newly added C within 1 year (Fig. 2). In comparison, 53.8% of the added new C entered and replenished the SOC stock. The greater magnitude of replenishment compared to the magnitude of the priming effect led to a net increase in SOC equivalent to 32.0% (25.3–38.8, a 95% confidence interval weighed with the variation and sample size in individual studies) of the added new C (Fig. 2).

**Model evaluation.** The priming effect and the replenishment in Fig. 2 were obtained from synthesis of individual isotope-labeled experiments, with a typical data set depicted in Fig. 3 (the example was from study 1 in Supplementary Data 1 and 2). New substrate addition significantly increased $CO_2$ emission from the old SOC (red dots), in comparison with that from the control (blue dots).

In the literature, the priming effect is traditionally expressed as a percent increase in $CO_2$ emission from the old SOC under the new substrate addition treatment relative to that in the control. For the case study in Fig. 3, the $CO_2$ emission from the old SOC in the substrate addition treatment was 34% greater than that in the control at the end of the incubation experiment (i.e., day 66; Fig. 3). The 34% increase in C loss from the old SOC was equivalent to 37.1% of the added $1000\,mg\,C\,g^{-1}$ soil. After 66 days of incubation, 40.2% of the added C was replenished to the soil C pools while 59.8% was directly released from the new substrate via microbial respiration, leading to a net SOC increase of 3.1% of the added C (Supplementary Data 2). In the current study, we quantified the priming effect relative to the amount of added C at standardized time points (Supplementary Fig. 1) in order to synthesize results from a group of diverse studies.

The four types of models described in Fig. 1 were trained by the data set in order to estimate the priming effect and replenishment at the end of 1 year. For the example shown in Fig. 3, the conventional model fitted the data extremely well in both the substrate addition

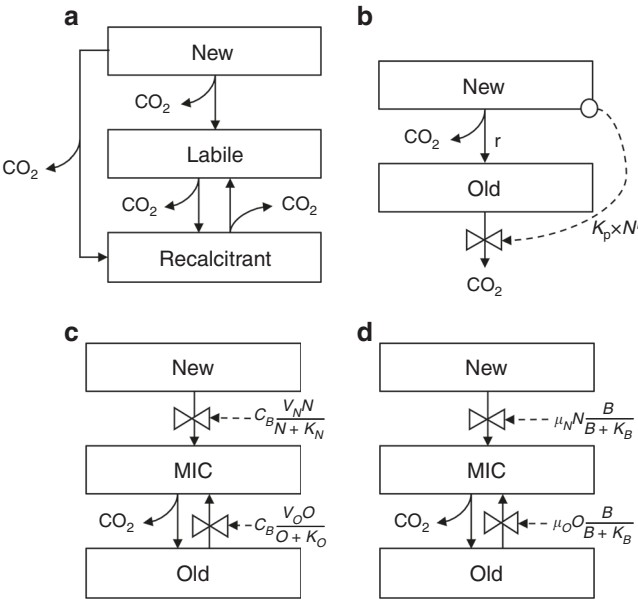

**Fig. 1** Schemes of four soil C dynamic models. **a** Conventional model; **b** Interactive model; **c** Michaelis–Menten model; **d** Reverse Michaelis–Menten model

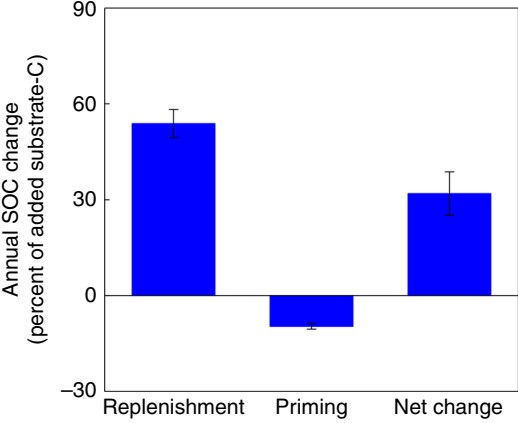

**Fig. 2** Synthesis of annual SOC change induced by replenishment and priming, and the consequent net SOC change with a one-time new C addition at the beginning. The magnitude of replenishment is significantly greater than that of priming, resulting in a net SOC accumulation. Mean ± 95% confidence interval

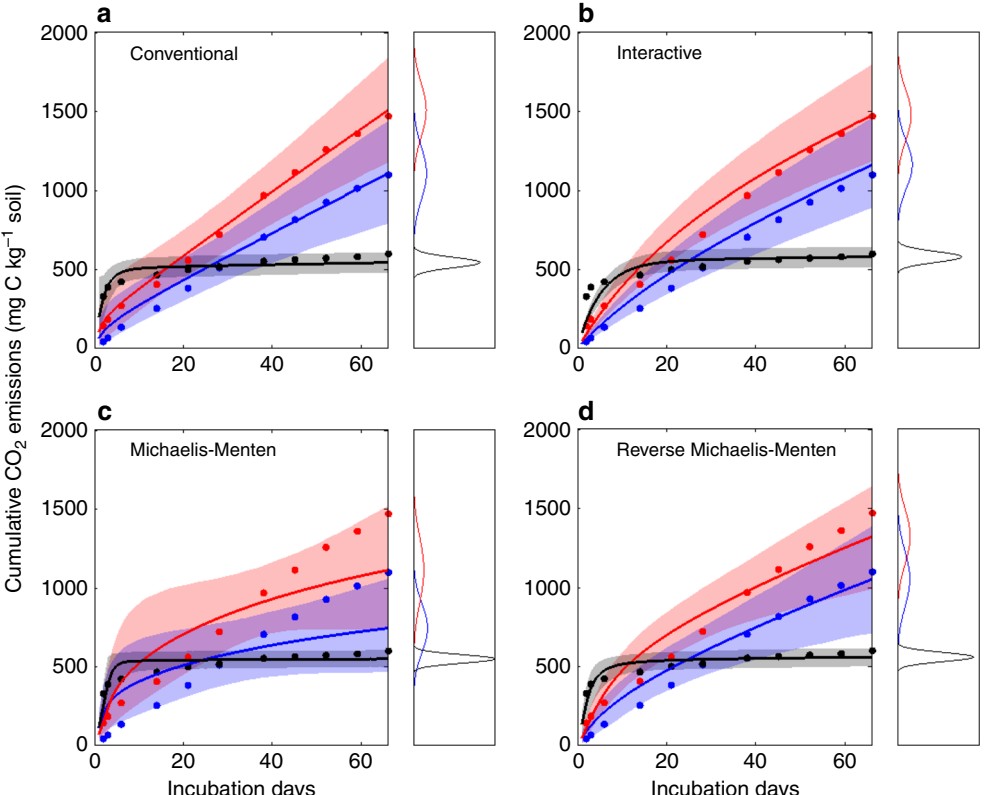

**Fig. 3** An example showing the performances of different models in simulating cumulative $CO_2$ emissions from old and new C substrates. Dots and lines are observations and model simulations, respectively. Shading areas are the simulated ranges from 2.5th to 97.5th percentiles (i.e., 95% range). Blue and red, $CO_2$ emissions from old C at the control and new C addition treatments, respectively; Black, $CO_2$ emissions from added new C. The distributions of model-simulated cumulative $CO_2$ emissions at the end of experiment are also shown in each panel

and control treatments separately ($R^2 = 0.99$; Fig. 3a and Supplementary Data 3). The interactive model performed very well for both the substrate addition and control treatments together ($R^2 = 0.97$; Fig. 3b and Supplementary Data 4). However, the regular Michaelis–Menten model tended to underestimate the cumulative $CO_2$ emissions from SOC for both the substrate addition and control treatments, especially toward the end of the experiment ($R^2 = 0.85$; Fig. 3c and Supplementary Data 5). Lastly, the reverse Michaelis–Menten model fitted the cumulative $CO_2$ emission data from the new substrate and old SOC in the control but not in the substrate addition treatment ($R^2 = 0.95$; Fig. 3d and Supplementary Data 6).

The model evaluation against all the training data (group I in Supplementary Data 1; statistically called within-sample evaluation) indicated that the regular Michaelis–Menten model did not adequately describe the SOC decomposition with priming and replenishment, showing a relatively high deviance information criterion (DIC) and a low data–model agreement (Fig. 4c and Table 1). Although the conventional and reverse Michaelis–Menten models reasonably fitted the cumulative $CO_2$ emissions, they did not demonstrate a high likelihood («0.5) of representation for replenishment and priming due to overfitting issues (Fig. 4a, d and Table 1). The interactive model fitted the data well, was the most parsimonious model (with the smallest DIC; Fig. 4b and Table 1), and was further validated by two modes with either fixed or randomized parameters (i.e., statistically called out-of-sample validation; Supplementary Figs. 2–4). The validation with either fixed or randomized parameters indicated that the calibrated interactive model well represents the priming effects regardless of experimental conditions. As such, the optimized interactive model was used to

estimate annual C replenishment, priming, and net SOC change across all studies (Fig. 2). In addition, we standardized the results by the amount of added new C.

**Impacts of abiotic and biotic factors.** After the standardization, neither the loss of the added C, nor replenishment, nor the net SOC change was dependent on the amount of added new C (Supplementary Fig. 5). In the experiments synthesized in this study, the soil water content ranged from 45 to 70% of water holding capacity and incubation temperatures varied from 0 to 28 °C. The estimated net C change was not dependent on either the soil water content or incubation temperature (Supplementary Fig. 6). In addition, the loss of the added C was not influenced by priming (Supplementary Fig. 7). After dividing the added C substrates into three categories: without N (e.g., glucose and starch), low N:C ratio (i.e., straw), and high N:C ratio (i.e., leaf materials), results showed that replenishment increased while priming decreased with an increase in substrate N:C ratio (Fig. 5). In addition, the net SOC increases in response to new C input ranged from 54 (high N:C ratio) to 41% (low N:C ratio) and to 19% (without N) of the newly added C (Fig. 5).

**Modeling experiment with continuous C inputs.** The synthesis conducted in this study was from data with a single C input at the beginning of experiments. Here we explored the effect of continuously increased new C input on SOC change using the interactive model. Results showed that a 10% step increase in C input starting from the beginning of the modeling experiment enhanced SOC by 43.1% of the total increased C input after 1 year (Fig. 6a). The increase in SOC induced by a gradual increase in C inputs was

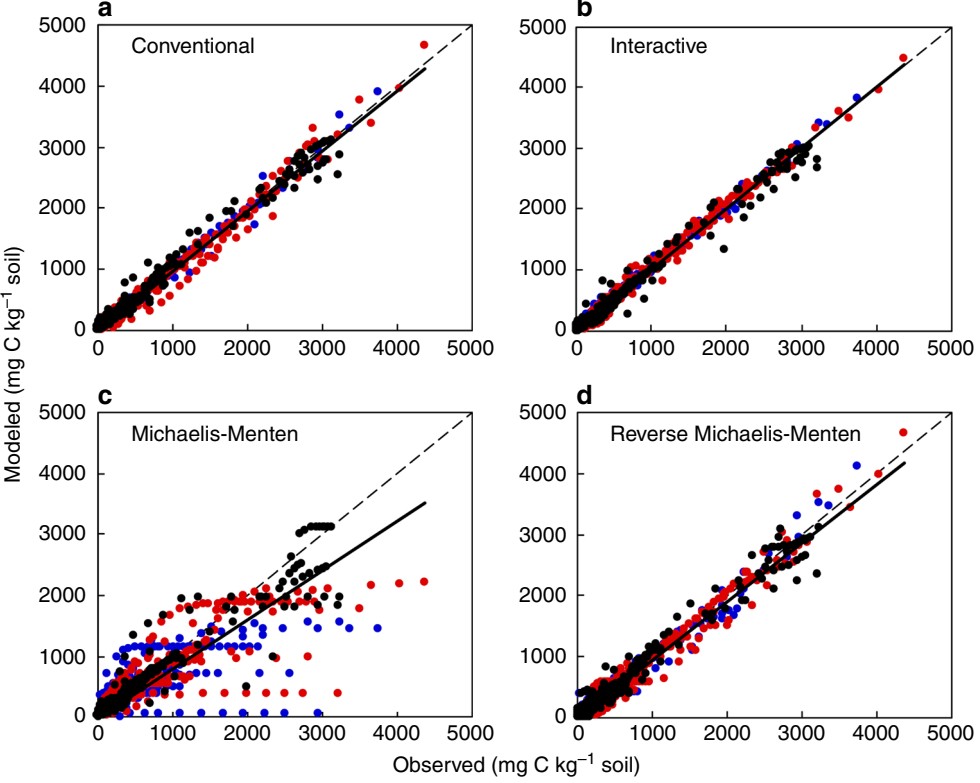

**Fig. 4** Within-sample model evaluation. Blue and red dots are $CO_2$ emissions from old C at the control and new C addition treatments, respectively; Black dots are $CO_2$ emissions from added new C; Solid line: linear regression (slope, $R^2$, $P$ values are shown in Table 1); Dashed line, 1:1 line

**Table 1 Performance of models in simulating SOC dynamics with replenishment and priming**

| Model | Number of parameters | Slope | $R^2$ | $P$ | DIC | Likelihood of model |
|---|---|---|---|---|---|---|
| Conventional | 12 | 0.98 | 0.99 | <0.01 | 50.92 | <0.01 |
| Interactive | 6 | 1.00 | 0.99 | <0.01 | 16.66 | 1.00 |
| Michaelis–Menten | 8 | 0.80 | 0.82 | <0.01 | 30.58 | <0.01 |
| Reverse Michaelis–Menten | 7 | 0.96 | 0.97 | <0.01 | 18.47 | 0.41 |

Number of parameters, slope, $R^2$, and $P$ values for the linear regression in Fig. 4, deviance information criterion (DIC), and likelihood of the models given the data for the within-sample evaluation are shown

45.5% of the total increased C input over the year (Fig. 6b). In addition, the magnitude of SOC change generally increased with the substrate N:C ratio (Supplementary Fig. 8). Overall, the modeling experiment confirmed that increased new C inputs promote accumulation of added substrates, which was independent of temperature and moisture conditions (Supplementary Fig. 9).

## Discussion

The general C accumulation after the additional new C input may be due to both physiochemical and biological interactions. First, the added new C might be protected by direct physical and chemical bonding to the soil mineral complex[17–19]. Second, a proportion of the new C can be utilized to increase microbial biomass with a concomitant production of metabolic by-products[12,13]. Through microbial processes the added C can also be transferred to the stable SOC fraction[12–14,19,20]. Although increased microbial growth may promote the decomposition of old SOC for energy and nutrient acquisition[4], our results illustrated that the amount of C loss resulting from the priming effect was five times smaller, on average, than the amount of replenished soil C. Despite the general pattern of C accumulations

following a new C input (Fig. 2), several individual studies have shown net SOC loss, primarily from saline alkaline[21] or low-fertility soils[22].

The enhanced replenishment with high substrate N:C ratio (Fig. 5) may be due to a more efficient utilization of substrates with high N:C ratios for the growth of microbial biomass compared to low N:C substrates[13,14,20,23]. In contrast, a higher priming loss of old SOC occurred when the added substrates have lower N:C ratios (Fig. 5), likely due to the scarcity of N. In this case, soil microbes scavenge N via the decomposition of old soil organic matter, resulting in stronger priming effects[3,24]. To further confirm the N mining hypothesis, we need more innovative incubation experiments to simultaneously quantify both C loss and N mineralization in response to additions of new C with different N content. These results suggest that the priming effect appears to become stronger, whereas the net increase in SOC resulting from the enhanced substrate input may decrease[9,25] if atmospheric $CO_2$ enrichment reduces the plant tissue N:C ratio[26–28].

In this study, the quantitative estimations were based on laboratory incubation experiments, which may be biased when applying in the field due to at least the two following reasons. First, disturbance and microenvironmental changes in the

incubation experiments may influence the magnitudes of the replenishment, priming, and net effect. Second, soil microbial community in the incubation jars may be different from that in the field. Thus, the values of the replenishment, priming, and net effect reported in this study should be used with caution.

Overall, this study quantitatively synthesized two important processes, replenishment and priming, in SOC dynamics. With the increase in C inputs, the magnitude of replenishment is generally greater than that of priming, resulting in net SOC accumulations over time. The results indicate that the anticipated increase in C inputs to soils under elevated $CO_2$ has the potential to mitigate climate change.

We also selected a best-fit model (i.e., the interactive model) using the extensive data sets. The interactive model, which represents priming by a power function of old C decomposition rate with the amount of the new C, is the most parsimonious. Our validation of the model with either fixed or randomized parameters indicates that the interactive model is able to well represent the priming effect and replenishment regardless of experimental conditions.

## Methods

**Data collection**. A comprehensive literature search, with key words "isotope" and "soil incubation", was conducted using the online search connection Web of Science in endnote. Three criteria were used to select the searched studies. First, the experiments included both the control and isotope-labeled C addition treatments. Second, SOC content, the added new C amounts, and multiple $CO_2$ emission rates (>2 time points) from old SOC and new substrates were reported. Third, the experiments lasted at least 4 weeks. Based on the criteria, 84 data sets from 26 publications were selected (Supplementary Data 1 and 2; refs. [21,22,29–52]). In addition, to explore the influence of substrate N:C ratio, the collected studies were divided into three groups: without N (e.g., glucose and starch), low N:C ratio (i.e., straw), and high N:C ratio (leaf materials). The amount of added C in most (i.e., over 2/3) of the collected studies fell within the range of <10% of SOC stocks. Global litter productivity is about 3–5% of global SOC stocks. The total C input to soils would be even more if considering root exudates though the global estimate is uncertain to our knowledge. In addition, Earth system models generally predict the C input could increase by 25–60% by the end of the twenty-first century[53]. Thus, the experimental additions are generally in accordance with the global C input estimates.

**Models**. Four different types of models, which had their respective assumptions to represent the replenishment and priming effect, were evaluated (Fig. 1). The four models included a conventional model, an interactive model, a regular Michaelis–Menten model, and a reverse Michaelis–Menten model. In the models, old C pools were those pre-existing and relative stable SOC, and new C pools were freshly added C which can be transferred to old C pools as decomposition proceeded. Like most Earth system models[54,55], the conventional model used first-order equations as shown below:

$$\frac{d\,^1N}{dt} = I - K_N \times {}^1N \tag{1}$$

$$\frac{d\,^1L}{dt} = K_N \times {}^1N \times a_{L,N} + K_R \times {}^1R \times a_{L,R} - K_L \times {}^1L \tag{2}$$

$$\frac{d\,^1R}{dt} = K_N \times {}^1N \times a_{R,N} + K_L \times {}^1L \times a_{R,L} - K_R \times {}^1R \tag{3}$$

$$\frac{d\,^1O}{dt} = \frac{d\,^1L}{dt} + \frac{d\,^1R}{dt} \tag{4}$$

$$\frac{d\,^nL}{dt} = K_N \times {}^nN \times a_{L,N} + K_R \times {}^nR \times a_{L,R} - K_L \times {}^nL \tag{5}$$

$$\frac{d\,^nR}{dt} = K_N \times {}^nN \times a_{R,N} + K_L \times {}^nL \times a_{R,L} - K_R \times {}^nR \tag{6}$$

$$\frac{d\,^nO}{dt} = \frac{d\,^nL}{dt} + \frac{d\,^nR}{dt} \tag{7}$$

$$\frac{dO}{dt} = \frac{d\,^1O}{dt} + \frac{d\,^nO}{dt} \tag{8}$$

$$f_L = \frac{{}^nL_0}{{}^nL_0 + {}^nR_0} \tag{9}$$

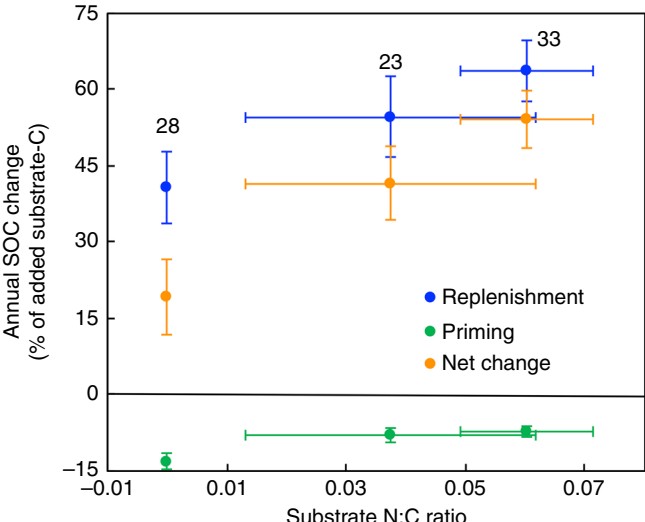

**Fig. 5** Synthesis of the dependence of annual replenishment, priming, and net SOC change on substrate N:C ratio. The replenishment increased, but priming decreased, with the increase in substrate N:C ratio. Thus, the net SOC change significantly increased with the increase in substrate N:C ratio. The number of studies for each category is shown near the bar. Mean ± 95% confidence interval

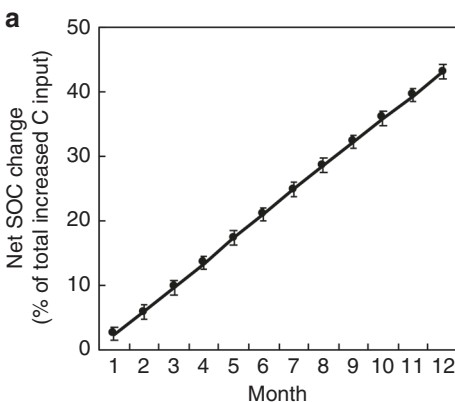

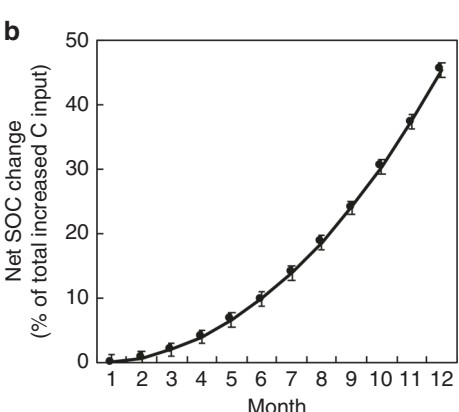

**Fig. 6** Synthesis of the modeling experiment. The modeling experiment showing net SOC increase by continuously increased C inputs. **a** Predicted net SOC change by a 10% step increase in C input for 1 year. **b** Predicted net SOC change by gradual increase in C input. Mean ± 95% confidence interval

where left superscript l and n mean isotope-labeled and non-labeled C pools. $N$ is newly added C pool. $L$ and $R$ are labile and recalcitrant pools, respectively. $I$ is new C input rate (mg C g$^{-1}$ soil per day), which is the amount of added substrate C in the substrate addition treatment at time 0. In the control treatment, $I$ is 0. $K_N$, $K_L$, and $K_R$ are decay rates (per day) of new C, labile SOC, and recalcitrant SOC; $a_{L,N}$ and $a_{R,N}$ are transfer coefficients (or carbon use efficiency, unitless) from new to labile and recalcitrant SOC, respectively; $a_{R,L}$ is transfer coefficient (unitless) from labile to recalcitrant SOC; $a_{L,R}$ is transfer coefficient (unitless) from recalcitrant to labile SOC; $O$ is the old SOC content (mg C g$^{-1}$ soil); $f_L$ is the initial fraction of the labile pool. The conventional model had two sets of parameters, as parameter changes were necessary to represent nonlinear processes in this type of models[9,56].

Based on the Introductory C Balance Model (ICBM)[57], the interactive model added a term to represent the priming effect. With the interactive model, the soil C dynamics were described as:

$$\frac{\mathrm{d}\,^lN}{\mathrm{d}t} = I - K_N \times \,^lN \tag{10}$$

$$\frac{\mathrm{d}\,^lO}{\mathrm{d}t} = K_N \times \,^lN \times r - (K_O + K_p \times N^p) \times \,^lO \tag{11}$$

$$\frac{\mathrm{d}\,^nN}{\mathrm{d}t} = -K_N \times \,^nN \tag{12}$$

$$\frac{\mathrm{d}\,^nO}{\mathrm{d}t} = K_N \times \,^nN \times r - (K_O + K_p \times N^p) \times \,^nO \tag{13}$$

$$\frac{\mathrm{d}N}{\mathrm{d}t} = \frac{\mathrm{d}\,^lN}{\mathrm{d}t} + \frac{\mathrm{d}\,^nN}{\mathrm{d}t} \tag{14}$$

$$\frac{\mathrm{d}O}{\mathrm{d}t} = \frac{\mathrm{d}\,^lO}{\mathrm{d}t} + \frac{\mathrm{d}\,^nO}{\mathrm{d}t} \tag{15}$$

$$f_N = \frac{^nN_0}{^nN_0 + \,^nO_0} \tag{16}$$

where $N$ and $O$ are the new and old C pools (mg C g$^{-1}$ soil); Correspondingly, $K_N$ and $K_O$ are the base decay rates (per day) of the two pools; $r$ and $K_p$ are the replenishment coefficient and priming coefficient (per day), respectively; $p$ is a factor (unitless) to determine the priming magnitude; $f_N$ is the the new pool fraction at the beginning.

The third model was the regular Michaelis–Menten model[58–60] (Fig. 1c) with the following equations:

$$\frac{\mathrm{d}\,^lN}{\mathrm{d}t} = I - B \times \frac{V_N \times \,^lN}{N + K_N} \tag{17}$$

$$\frac{\mathrm{d}\,^lO}{\mathrm{d}t} = \mu_B \times B - B \times \frac{V_O \times \,^lO}{O + K_O} \tag{18}$$

$$\frac{\mathrm{d}\,^lB}{\mathrm{d}t} = -\mu_B \times \,^lB + \varepsilon \times \,^lB \times \left( \frac{V_N \times N}{N + K_N} + \frac{V_O \times O}{O + K_O} \right) \tag{19}$$

$$\frac{\mathrm{d}\,^nN}{\mathrm{d}t} = -B \times \frac{V_N \times \,^nN}{N + K_N} \tag{20}$$

$$\frac{\mathrm{d}\,^nO}{\mathrm{d}t} = \mu_B \times B - B \times \frac{V_O \times \,^nO}{O + K_O} \tag{21}$$

$$\frac{\mathrm{d}\,^nB}{\mathrm{d}t} = -\mu_B \times \,^nB + \varepsilon \times \,^nB \times \left( \frac{V_N \times N}{N + K_N} + \frac{V_O \times O}{O + K_O} \right) \tag{22}$$

$$\frac{\mathrm{d}N}{\mathrm{d}t} = \frac{\mathrm{d}\,^lN}{\mathrm{d}t} + \frac{\mathrm{d}\,^nN}{\mathrm{d}t} \tag{23}$$

$$\frac{\mathrm{d}O}{\mathrm{d}t} = \frac{\mathrm{d}\,^lO}{\mathrm{d}t} + \frac{\mathrm{d}\,^nO}{\mathrm{d}t} \tag{24}$$

$$\frac{\mathrm{d}B}{\mathrm{d}t} = \frac{\mathrm{d}\,^lB}{\mathrm{d}t} + \frac{\mathrm{d}\,^nB}{\mathrm{d}t} \tag{25}$$

$$f_N = \frac{^nN_0}{^nN_0 + \,^nB + \,^nO_0} \tag{26}$$

$$f_B = \frac{^nB_0}{^nN_0 + \,^nB + \,^nO_0} \tag{27}$$

where $N$, $O$, and $B$ are pool sizes (mg C g$^{-1}$ soil) of new C, old C, and microbial biomass; $V_N$ and $V_O$ are maximum substrate C (new or old C) assimilation rates (per day); $K_N$ and $K_O$ are Michaelis–Menten constants (mg C g$^{-1}$ soil); $\mu_B$ is turnover rate of microbial biomass (per day); $\varepsilon$ is microbial growth efficiency (unitless); $f_N$ and $f_B$ are the initial fraction of the new and microbial biomass pools.

The fourth model was the reverse Michaelis–Menten model[61, 62] as,

$$\frac{d\,^1N}{dt} = I - \mu_N \times {}^1N \times \frac{B}{B + K_B} \tag{28}$$

$$\frac{d\,^1O}{dt} = \mu_B \times B - \mu_O \times {}^1O \times \frac{B}{B + K_B} \tag{29}$$

$$\frac{d\,^1B}{dt} = -\mu_B \times {}^1B + \varepsilon \times (\mu_N \times N + \mu_O \times O) \times \frac{^1B}{B + K_B} \tag{30}$$

$$\frac{d\,^nN}{dt} = -\mu_N \times {}^nN \times \frac{B}{B + K_B} \tag{31}$$

$$\frac{d\,^nO}{dt} = \mu_B \times B - \mu_O \times {}^nO \times \frac{B}{B + K_B} \tag{32}$$

$$\frac{d\,^nB}{dt} = -\mu_B \times {}^nB + \varepsilon \times (\mu_N \times N + \mu_O \times O) \times \frac{^nB}{B + K_B} \tag{33}$$

$$\frac{dN}{dt} = \frac{d\,^1N}{dt} + \frac{d\,^nN}{dt} \tag{34}$$

$$\frac{dO}{dt} = \frac{d\,^1O}{dt} + \frac{d\,^nO}{dt} \tag{35}$$

$$\frac{dB}{dt} = \frac{d\,^1B}{dt} + \frac{d\,^nB}{dt} \tag{36}$$

$$f_N = \frac{^nN_0}{^nN_0 + {}^nB + {}^nO_0} \tag{37}$$

$$f_B = \frac{^nB_0}{^nN_0 + {}^nB + {}^nO_0} \tag{38}$$

where $N$, $O$, and $B$ are pool sizes (mg C g$^{-1}$ soil) of new C, old C, and microbial biomass; $\mu_N$, $\mu_O$, and $\mu_B$ are turnover rates (per day) of new C, old C, and microbial biomass; $K_B$ is a coefficient for C consumption by microbes (mg C g$^{-1}$ soil); $\varepsilon$ is microbial growth efficiency (unitless); $f_N$ and $f_B$ are the initial fraction of the new and microbial biomass pools.

**Model optimization and selection.** The model optimization was based on Bayes' theorem:

$$P(\theta|Z) \propto P(Z|\theta)P(\theta) \tag{39}$$

where $P(\theta)$ and $P(\theta|Z)$ of model parameters ($\theta$) are the priori and posterior probability density function (PDF), respectively. Uniform distributions over parameter ranges were used as the priori PDFs. $P(Z|\theta)$ is the likelihood function of data, which was calculated as:

$$P(Z|\theta) \propto \exp\left\{ -\sum_{i=1}^{n} \sum_{t \in \mathrm{obs}(Z_i)} \frac{[Z_i(t) - X_i(t)]^2}{2\sigma_i^2(t)} \right\} \tag{40}$$

where $Z_i(t)$ and $X_i(t)$ are the observed and modeled values, respectively. $\sigma_i(t)$ is the standard deviation of measurements.

The adaptive Metropolis–Hastings algorithm was used to optimize the model parameters for each study[63,64]. The algorithm included two primary steps[54,65]: First, a new random value ($\theta^{\mathrm{new}}$) was generated from the accepted value of the previous step ($\theta^{\mathrm{old}}$):

$$\theta^{\mathrm{new}} = \theta^{\mathrm{old}} + d(\theta_{\max} - \theta_{\min})/D \tag{41}$$

where $\theta_{\max}$ and $\theta_{\min}$ are the priori PDF boundaries, $D$ is step size, and $d$ is randomly selected between $-0.5$ and $0.5$. Second, $\theta^{\mathrm{new}}$ was tested against the Metropolis criterion to accept or reject. The two steps were repeated to generate the posterior PDFs of parameters, after discarding the first half of accepted values. The maximum likelihood estimates (MLEs) of the parameters of the four models are shown in Supplementary Data 3–6.

Deviance information criterion (DIC)[16] and likelihood of model[66] were used to evaluate the models given the data. For each study, DIC was calculated by

$$\mathrm{DIC} = \bar{D} + p_D \tag{42}$$

where

$$\bar{D} = \frac{1}{S} \sum_{i=1}^{S} \left(-2\log\left(P(Z|\theta^i)\right)\right) \tag{43}$$

and

$$p_D = \bar{D} + 2\log\left(P(Z|\bar{\theta})\right) \tag{44}$$

where $S$ is the number of the generated parameter sets, and $\bar{\theta}$ is the mean of the generated parameter sets. The weighted average DIC for all studies was calculated by

$$\mathrm{DIC_w} = \frac{\sum_{i=1}^{71} \mathrm{DIC}_i N_i}{\sum_{i=1}^{71} N_i} \tag{45}$$

where $N_i$ is the number of data points in the $i$th study. The smaller DIC is for a model, the better the model is. The likelihood ($L$) of model given the data was calculated by

$$L = e^{-0.5(\mathrm{DIC_w} - \mathrm{DIC_{min}})} \tag{46}$$

where $\mathrm{DIC_{min}}$ is the minimum $\mathrm{DIC_w}$ value of the four models. In this study, 0.5 was used as a threshold for $L$ to select model. Only the interactive model had a $L$ value bigger than 0.5 (Table 1).

**Model validation.** To further validate the selected model (i.e., interactive model in this study) as an assurance of model extrapolation beyond the observations, we employed two modes with both fixed and random parameters for model validation. In the fixed mode of validation, we used three collected publications in which different amounts of new C were added into the same soils[32,40,42]. For those studies, one new C amount was used for model selection and parameter optimization (studies 8, 34, 36, and 42 in Supplementary Data 1; training group), and the others were used for model validation (studies 9, 10, 35, 37, and 43 in Supplementary Data 1; validation group). The interactive model with optimized (i.e., fixed) parameters with the training data was run with the new C amount at the validation group, and the modeled decomposition rates of SOC and added substrates were compared with observed ones (Supplementary Fig. 2). In the random mode of model validation, the collected 84 data sets were randomly divided into two groups, one for model training and the other for validation. The two groups had similar distribution of the added new C amount (% of SOC) (Supplementary Fig. 3). The trained interactive model by the first group of data was run to predict the priming effect of the second group of data, and was compared with the observations (Supplementary Fig. 4). The model selection and validation results indicated that the selected interactive model had the ability to represent SOC decomposition with priming and replenishment. Thus, the interactive model was used for further analyses.

**Estimation of C fluxes.** At each time step, the replenishment is calculated as the amount of isotope-labeled C left in the soil C pools after microbial respiration. The priming effect is the difference of cumulative $CO_2$ emission from non-labeled old SOC between the substrate addition treatment and the control. The net effect of the substrate addition on SOC is the difference of total SOC between the substrate addition treatment and the control and can be calculated by subtracting the priming effect from the replenishment.

To synthesize results from a group of diverse studies, the estimations were normalized relative to the amount of added C at a standardized time point. The C dynamics at 1 year after the incubation are at a relatively stable phase (Supplementary Fig. 1). Therefore, the results derived from the modeled C dynamics at 1 year are robust to estimate annual replenishment, priming, and SOC content.

In addtion to the annual estimation, we also conducted a modeling experiment to reveal the effect of continuously increased new C input on SOC change. For each of the data sets with a diverse range of different soils and substrate quality, the optimized interactive model was spun up to steady state with the C input of 0.8 mg C kg$^{-1}$ soil per day, which approximates a global average C input of 378 g m$^{-2}$ per year to the topsoil (1 m) with an assumed soil bulk density of 1.3 g cm$^{-3}$ (ref. [67]). After reaching steady state, the model was run for 365 days under three C input scenarios. The first and the second scenarios had constant C inputs of 0.80 (i.e., no increase in C input) and 0.88 (i.e., a 10% step increment) mg C kg$^{-1}$ soil per day, respectively. In the third scenario, the C input increased linearly from 0.8004 mg C kg$^{-1}$ soil per day in the first day to 0.9596 mg C kg$^{-1}$ soil per day in the 365th day. The total C input in the latter two scenarios was 29.2 mg C kg$^{-1}$ soil greater than that in the first scenario over the year. In addition, we manipulated incubation temperature and moisture to reveal the effect of those environmental factors on the SOC change. To explore the warming effect, we increased temperature by 2 °C. Assuming the temperature sensitivity ($Q_{10}$) was 2, the warming manipulation increased the rate parameters ($k_1$, $k_2$, and $k_p$) by 15%. By assuming a linear moisture limitation, we also conducted a drying and a wetting treatment, in which the rate parameters were decreased and increased by 10%, respectively.

**Synthesis**. To estimate the weighted mean ($M_{weighted}$) and 95% confidence interval (CI), we treated the estimated priming effect, replenishment, or net SOC change, respectively, as one replicate for each study. The weighted mean and 95% CI were estimated in the MetaWin 2.1 using a meta-analysis technique[68,69]. Briefly, the weighted mean was calculated as

$$M_{\text{weighted}} = \frac{\sum_{j=1}^{k} W_j^* M_j}{\sum_{j=1}^{k} W_j^*} \qquad (47)$$

with the variance as

$$V_{\text{weighted}} = \frac{1}{\sum_{j=1}^{k} W_j^*} \qquad (48)$$

where $M_j$ is the mean value of study $j$, $W_j^*$ is the weighting factor. The 95% confidence interval (CI) of the weighted mean was estimated as

$$\text{CI} = M_{\text{weighted}} \pm 1.96 \times \sqrt{V_{\text{weighted}}} \qquad (49)$$

All the results were tested at $P < 0.05$ level based on the 95% CI. If the 95% CI did not overlap with zero, it was considered statistically significant. Similarly, non-overlapped 95% CIs indicated significant differences among groups.

**Code availability**. The computer code used to run the simulations is available upon request.

**Data availability**. The data used can be found in Supplementary Information.

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

## Acknowledgements

This study was financially supported by the US Department of Energy (DOE), Office of Biological and Environmental Research, Terrestrial Ecosystem Sciences grant DE SC00114085, Biological Systems Research on the Role of Microbial Communities in Carbon Cycling Program grants DE-SC0004601 and DE-SC0010715, the Terrestrial Ecosystem Science Scientific Focus Area (TES-SFA) at Oak Ridge National Laboratory (ORNL), and US National Science Foundation (NSF) grants EF 1137293 and OIA-1301789. ORNL is managed by the University of Tennessee-Battelle, LLC, under contract DE-AC05-00OR22725 with the US DOE.

## Author contributions

J.L. and Y.L. designed the study. J.L., X.L., and C.H. collected and organized the data. J.L. built the model. J.L., Z.Z., L.H., and B.L. performed the analyses. J.L. and Y.L. wrote the first draft of the paper. J.L., Z.S., J.R.C., K.T.K., Z.L, C.R.P., E.A.G.S., J.M.T., Y.-P.W., L.W., J.X., J.Z., and Y.L. contributed to discussing the results and manuscript revisions.

## Additional information

**Competing interests:** The authors declare no competing interests.

