## [Peer Review File · Nature Communications]

Reviewers' comments:

Reviewer #1 (Remarks to the Author):

Review of "More Replenishment than Priming Loss of Soil Organic Carbon with Additional Carbon Input" by Liang et al.

This is a very interesting study that examined a critical question in the soil science community, especially under the warming climate: what is the relative contribution of increased litter and rhizosphere C that replenishes soil C stock, versus the loss of old C due to priming effect. This study synthesized 76 datasets and showed that C replenish outweigh C loss, and that the net gain is correlated with N/C ratio of the added substrates.

I have the following specific comments:

1. How is potential microbial community or dynamics shift accounted for in this study? Some microbial based soil models project net loss of carbon under increased above-ground inputs. Ah, I see you did include microbial models in your analysis! But maybe it is good to mention this in the abstract so reader can easily gauge what have/have not been considered.
2. It is usually assumed that priming effect only affects old C, however it might also impact the new C (added C), how is this assumption validated in the study?
3. I see the dataset are separated into training vs validation set, how did you make sure that the distribution of the two datasets are alike, or did you examine the similarity of the two sets? Validation is only effective when the training and validation sets are from the same distribution to avoid biases
4. L130-150: when talk about model fitting performance, it would be good to mention also some statistics rather than just pure narrative description (e.g. "performed very well", but by how much?), so that reader will have a quantitative sense. Also, are these fitting performance based on the validation set? It should be out-of-sample test. If a model is not able to capture the patterns from the synthesized data, its extrapolation is likely to be questionable as well.
5. The collected studies are all short-term, with max length of 168 days, about half a year. How do you justify that this is sufficient for informing models?

Reviewer #2 (Remarks to the Author):

This study is timely and the topic is of general scientific interest. Mechanisms regulating soil C stocks and how best to predict soil C stocks in a changing climate is continually being refined. The overall rationale for this study is strong, but the broad conclusions made in this paper are not necessarily convincing as presented.

The study addresses how additional C inputs enhance the decomposition and loss of old soil C (priming). The authors synthesize results from several isotope-tracing laboratory experiments that explicitly define soil organic matter priming and the fate of new C inputs. The study expands on these results, using a data-model synthesis exercise. They conclude that new C inputs will result in a net increase in soil C and the magnitude of this response is largely driven by the C:N ratio of added substrates. They then use a 100-year modeling experiment to show that new C inputs lead to significant soil C accumulation.

The claims made by the authors are perhaps over-reaching as they are based on extrapolating short-term laboratory data to long-term ecosystem effects, without providing a clear description on how the climatic and ecosystem complexities were accounted for. The authors further suggest that the best-fit model from laboratory data synthesis should be used for ecosystem and earth system models, although the study lacks a highly-convincing argument for the application at broad scales. The lack of detail provided for such extrapolation make it difficult to evaluate these claims.

I detail specific aspects I find problematic below.

Specific Concerns:

Lines 84-87. How realistic were these substrate additions? How do they relate to productivity inputs or SOC stock? It appears that the range in substrate additions is quite large (from <1% to nearly 35% of SOC stocks). Wouldn't the amount have significant influence on the fate of the added substrate? How might this be influencing the results since replenishment was defined as the amount of substrate C remaining? Additionally, how are temperature and soil moisture being accounted for? Were all incubations performed at the same temperature and moisture? If not, how was this included in the analysis described in this synthesis?

Lines 84-87. Model evaluation was performed with the majority of the data in this dataset (71 of the 76 studies), while only 5 studies were used for model evaluation. It appears that the 5 data sets used for model evaluation came from only 3 studies. I am not convinced that 3 studies provide a robust model model evaluation, especially because it appears these data represent only 3 locations, soil types, climate zones, etc. Additionally, one of the referenced studies used in model evaluation performed the laboratory experiment for 209 days, while the authors state they only used studies that occurred 28-168 days. It is unclear if all the data was used or if only a certain amount of data was selected and why.

Lines 112-117. When evaluating the data set, it appears there is a highly significant linear relationship between net SOC change and the amount of C added. Thus, the replenishment effects are significantly influenced by the amount added. Was the amount added in these studies justified based on certain criteria? The authors do standardize the results by the amount of substrate added, but even within one soil type one may expect different responses based on the amount of substrate added. Could this be influencing the overall effect and conclusions made in this synthesis?

Lines 142-144. Unclear sentence. What is meant by "complex SOC"? It is unclear how the model inadequately described priming with complex SOC. Can complexity be ascribed to simple C:N ratios? What is meant by "mono-substrates"?

Lines 150-152. The claim that this particular model can be used for ecosystem and earth-system models is too far reaching. It is unclear how the data based on laboratory incubations is being extrapolated to the ecosystem. How are different climatic variables and soil properties being used? Would introducing this complexity strengthen or weaken these results?

Lines 163-180. There are a lot of details missing from the description of the 100-year modeling experiment. First, I am not convinced that that the laboratory data is appropriately used to describe long-term dynamics. There are a number of studies that show how temperature and soil moisture can significantly influence the magnitude of soil C loss. How was this varied in the 100-year modeling exercise? What about N?

Lines 182-186. Why not explore these mechanisms further? Could the range of soil and climatic characteristics from this data set be used to explore the factors regulating SOC dynamics?

Lines 196-200. Why not expand on this? If the studies also measured soil nutrients you should see this in the data set.

Lines 213-216. This conclusion is a stretch and I am not convinced based on this study one can conclude what is best for ecosystem and Earth system models.

Lines 330-332. '100 sets of parameters' - It is unclear what this means as written.

Supplementary Table 1: It would be informative to include the location from which these soils were sampled.

Reviewer #3 (Remarks to the Author):

The paper is an interesting study of carbon priming affects on soil organic carbon stocks, a topic of great relevance to earth system models. The study combines a meta-analysis of empirical studies from a broad range of systems in relation to four types of models that are commonly used to represent SOC dynamics. The study is significant for at least two reasons. First, the replenishment to priming ratios show a relatively narrow range despite the variation in soil types and added substrate composition with direct application. Second, the exercise shows that the most parsimonious model (interactive) offers the greatest predictive value. There are frequent calls in the literature for inclusion of more microbial detail in earth system models. This study illustrates that such detail, which may be difficult acquire, may not be helpful. The study is thorough and well described. It is difficult to digest given the need to keep referring to supplemental materials and the epilogue methods section for relevant context.

Line 159-165. The N:C result is consistent with the N mining hypothesis that labile carbon inputs stimulate the degradation of SOC for N acquisition. This topic is raised again in the discussion (line 199). This interaction of C priming and N mining is interesting. The authors might consider expanding on this topic. With the data on hand, it appears possible to estimate the potential N mineralization from SOC in relation to substrate N input. This exercise is analogous to the C input – C mineralization results presented, and might provide additional insight into the controls of SOM accumulation.

Reviewers' comments:

Reviewer #1 (Remarks to the Author):

Review of “More Replenishment than Priming Loss of Soil Organic Carbon with Additional Carbon Input” by Liang et al.

This is a very interesting study that examined a critical question in the soil science community, especially under the warming climate: what is the relative contribution of increased litter and rhizosphere C that replenishes soil C stock, versus the loss of old C due to priming effect. This study synthesized 76 datasets and showed that C replenish outweigh C loss, and that the net gain is correlated with N/C ratio of the added substrates.

I have the following specific comments:

1. How is potential microbial community or dynamics shift accounted for in this study? Some microbial based soil models project net loss of carbon under increased above-ground inputs. Ah, I see you did include microbial models in your analysis! But maybe it is good to mention this in the abstract so reader can easily gauge what have/have not been considered.

Response: We greatly appreciate the reviewer for this excellent suggestion. We added a sentence in the Abstract to describe the point on lines 58-60:

“Additionally, model evaluation indicated that a two-pool interactive model, instead of nonlinear models with explicit microbial pools, was the most parsimonious one to represent the SOC decomposition with priming and replenishment.”

2. It is usually assumed that priming effect only affects old C, however it might also impact the new C (added C), how is this assumption validated in the study?

Response: We appreciate the reviewer for pointing out this interesting issue. Priming by definition refers to microbial processes that promote its growth partially via liberation of old soil carbon. We are not aware of any studies in the literature to quantify impacts of priming on the newly added carbon. We truly believe your idea deserves careful thinking by experimentalists to design new experiments to explore. This study is a synthesis of published results in the literature and was not designed to mechanistically exam this issue. Validating this assumption is beyond the scope of this study. Nevertheless, we conducted one additional analysis to exam whether loss of newly added C is correlated with priming. Our analysis showed that the loss of the added C was not impacted by priming as shown below and in Supplementary Fig. 7 of the revised manuscript.

3. I see the dataset are separated into training vs validation set, how did you make sure that the distribution of the two datasets are alike, or did you examine the similarity of the two sets? Validation is only effective when the training and validation sets are from the same distribution to avoid biases

Response: The reviewer asked a very insightful question. In the last version, we used five datasets for validation, each was corresponding to at least one dataset for training. The paired datasets were collected from the same experiments. Thus, estimated parameters on priming from the training dataset can be evaluated with the validation dataset. In this sense, our previous validation used fixed parameters during the validation (or called it a fixed mode of validation). To broadly test whether the estimated priming can be validated by any independent experiments, we conducted another model validation (or called it a random mode of validation) in the revised manuscript. The collected data sets were randomly divided into two groups, one for model training and the other for validation. The two groups had similar distribution of the added new C amount (% of SOC) (Supplementary Figure 3 and below; panel a is for the training and panel b is for the validation).

The trained interactive model by the first group of data was run to predict the priming effect of the second group of studies, and was compared with the observations. The results showed that the model predictions and observations have very similar distributions (Supplementary Figure 4 and below; the black stairs and the red bars are modeled and observed, respectively), indicating that estimated priming with the interactive model well represented the SOC decomposition.

In the revised manuscript, we added sentences on lines 146-150 to describe the results as:

“The interactive model fitted data well, was the most parsimonious model (with the smallest DIC; Fig. 4b, e), and was further validated by two modes with either fixed or randomized parameters (i.e., statistically called out-of-sample validation; Supplementary Figs. 2-4). The validation with either fixed or randomized parameters indicated that the calibrated interactive model well represents the priming effects regardless of experimental conditions.”

We also added a new section under Methods to describe the details (lines 352-369) as

“Model validation. To further validate the selected model (i.e., interactive model in this study) as an assurance of model extrapolation beyond the observations, we employed two modes with both fixed and random parameters for model validation. In the fixed mode of validation, we used three collected publications in which different amounts of new C were added into the same soils¹⁻³. For those studies, one new C amount was used for model selection and parameter optimization (studies 8, 34, 36, 42 in Supplementary Table 1; training group), and the others were used for model validation (studies 9, 10, 35, 37, 43 in Supplementary Table 1; validation group). The interactive model with optimized (i.e., fixed) parameters with the training data was run with the new C amount at the validation group, and the modeled decomposition rates of SOC and added substrates were compared with observed ones (Supplementary Fig. 2). In the random mode of model validation, the collected 84 datasets were randomly divided into two groups, one for model training and the other for validation. The two groups had similar distribution of the added new C amount (% of SOC) (Supplementary Fig. 3). The trained interactive model by the first group of data was run to predict the priming effect of the second group of data, and was compared with the observations (Supplementary Fig. 4). The model selection and validation results indicated that the selected interactive model had the ability to represent SOC decomposition with priming and replenishment. Thus, the interactive model was used for further analyses.”

Technically, validation is not necessary when the data assimilation technique was used to select the most parsimonious model (i.e., the interactive model in this study) and estimate the optimal parameters to represent the SOC decomposition with priming and replenishment. The model validation with both the fixed and random modes serves as a double-assurance to strengthen our confidence of the conclusions.

4. L130-150: when talk about model fitting performance, it would be good to mention also some statistics rather than just pure narrative description (e.g. “performed very well”, but by how much?), so that reader will have a quantitative sense. Also, are these fitting performance based on the validation set? It should be out-of-sample test. If a model is not able to capture the patterns from the synthesized data, its extrapolation is likely to be questionable as well.

Response: In the manuscript, we described the model fitting performance using study 1 in Supplementary Table 1 and Supplementary Dataset 1 as an example. We used this example to provide a basic sense of data for the synthesis. In the revised manuscript, we added the coefficient of determination (R^2) to describe the results (lines 128-138).

In addition to this example, we provided the model performance in comparison with all the collected data (Fig. 4, Supplementary Figs. 2-4). We first conducted a within-sample evaluation to select the most parsimonious (i.e., complex enough to represent the SOC decomposition with priming and replenishment AND simple enough to avoid over-fitting in terms of the available data) model from four candidates (Fig. 1). In Fig. 4, we provided a series of statistical indices to select the model. Based on the deviance information criterion (DIC)⁴, we selected the interactive model (Fig. 1b) as the most parsimonious one. The selected interactive model matched the

observed data very well ($R^2 = 0.99$). More importantly, it was simple enough to avoid overfitting issue (with the smallest DIC and the highest likelihood of model; Fig. 4e) (lines 139-148).

Then, the selected interactive model was further evaluated by two modes of out-of-sample validation as described in the response to Comment # 3 of this reviewer (Supplementary Figs. 2-4). These results suggested that estimated C fluxes in this study can be extrapolated beyond the training group of data.

In summary, all the statistics for model selection and validation indicated that the selected interactive model has a high fidelity to represent the SOC decomposition with priming and replenishment, and the estimated SOC decomposition is sufficiently robust beyond the conditions in specific experiments from which data sets were collected.

5. The collected studies are all short-term, with max length of 168 days, about half a year. How do you justify that this is sufficient for informing models?

Response: This is an excellent question. Generally speaking, if a short-term study reveals fundamental processes that operate at different time scales, the results of such a study can be used to inform models. A classic example is the leaf photosynthesis model by Farquhar et al. (1980)⁵. Almost all the processes represented in Farquhar model are short-term, fast biochemical processes. Those processes were mostly studied in the laboratory. However, Farquhar model, which synthesizes fast biochemical processes from short-term experiments, has been used at time scales from minutes to centuries and at spatial scales from biochemistry to the globe.

If we believe those short-term experiments included in our study reveal fundamental processes underlying priming and replenishment, our synthesis of those experimental results into models shall be valid in term of representing priming and replenishment in the model. There are two questions to consider. First, what would happen if those experiments last much longer than several months? Most scientists did not do experiments much longer than several months likely because longer experiments would not reveal anything new. Second, can the interactive model be used to study priming and replenishment in the real world? Because the model performed quite well in simulating the collected data in this study, we can hypothesize the model should be able to present priming and replenishment in the real world. To test this hypothesis, scientists need to design new experiments and collect more data to either falsify or confirm the model.

To account for the concern on the point that the short-term laboratory incubation experiments may not be sufficient to inform long-term models, we removed the long-term simulation in the revised manuscript. Instead, we conducted a short-term (i.e., 1 year) modeling incubation experiment to analyze the effect of continuously increased C addition on the SOC change (Fig. 6 Supplementary Figs. 8-9; lines 165-172; lines 382-398). This short-term modeling experiment with continuous C input strengthened our conclusion that that increased new C input generally enhances SOC sequestration.

Reviewer #2 (Remarks to the Author):

This study is timely and the topic is of general scientific interest. Mechanisms regulating soil C stocks and how best to predict soil C stocks in a changing climate is continually being refined. The overall rationale for this study is strong, but the broad conclusions made in this paper are not necessarily convincing as presented.

The study addresses how additional C inputs enhance the decomposition and loss of old soil C (priming). The authors synthesize results from several isotope-tracing laboratory experiments that explicitly define soil organic matter priming and the fate of new C inputs. The study expands on these results, using a data-model synthesis exercise. They conclude that new C inputs will result in a net increase in soil C and the magnitude of this response is largely driven by the C:N ratio of added substrates. They then use a 100-year modeling experiment to show that new C inputs lead to significant soil C accumulation.

The claims made by the authors are perhaps over-reaching as they are based on extrapolating short-term laboratory data to long-term ecosystem effects, without providing a clear description on how the climatic and ecosystem complexities were accounted for. The authors further suggest that the best-fit model from laboratory data synthesis should be used for ecosystem and earth system models, although the study lacks a highly-convincing argument for the application at broad scales. The lack of detail provided for such extrapolation make it difficult to evaluate these claims. I detail specific aspects I find problematic below.

Response: We greatly appreciate the thoughtful comments. We welcome your critiques and have revised the manuscript accordingly. Please see the point-by-point responses below.

Specific Concerns:

Lines 84-87. How realistic were these substrate additions? How do they relate to productivity inputs or SOC stock? It appears that the range in substrate additions is quite large (from <1% to nearly 35% of SOC stocks). Wouldn't the amount have significant influence on the fate of the added substrate? How might this be influencing the results since replenishment was defined as the amount of substrate C remaining?

Response: We really appreciate the reviewer for asking those critical questions.

Reviewer's question "how realistic were these substrate additions?" may or may not be considered by scientists who conducted those original experiments. Fortunately, the wide "range in substrate additions (from <1% to nearly 35% of SOC stocks)" covers reality as the global litter productivity is about 3-5% of global SOC stocks. Thus, the results of our synthesis with the standardization by the amount of added C should be applicable to the real world.

To answer reviewer's question on the possible influence of the amount of added C on the fate of the added substrates, we conducted a further analysis. We plotted the loss of added C, replenishment and net SOC change against the added C amount (Supplementary Fig. 5). Results showed that the amount of added substrate did not have any significant effects on those variables as shown below and in Supplementary Fig. 5.

Additionally, how are temperature and soil moisture being accounted for? Were all incubations performed at the same temperature and moisture? If not, how was this included in the analysis described in this synthesis?

Response: Those are great questions. In those experiments synthesized in this study, the soil water content ranged from 45% to 70% of water holding capacity. Temperature varied from 0 to 28 °C. Our analyses indicated that neither incubation temperature nor moisture had any clear influences on the SOC change after new C addition as shown below and in Supplementary Fig. 6 of the revised manuscript.

Lines 84-87. Model evaluation was performed with the majority of the data in this dataset (71 of the 76 studies), while only 5 studies were used for model evaluation. It appears that the 5 data sets used for model evaluation came from only 3 studies. I am not convinced that 3 studies provide a robust model model evaluation, especially because it appears these data represent only 3 locations, soil types, climate zones, etc. Additionally, one of the referenced studies used in model evaluation performed the laboratory experiment for 209 days, while the authors state they only used studies that occurred 28-168 days. It is unclear if all the data was used or if only a certain amount of data was selected and why.

Response: This reviewer asked a very important question. Reviewer 1 also asked a similar question (i.e., question 3). We responded to question 3 of reviewer 1 above.

Technically, validation is not necessary when the data assimilation technique was used to select the most parsimonious model (i.e., the interactive model in this study) and estimate the optimal parameters to represent the SOC decomposition with priming.

In the last version, we used five datasets for validation, each was corresponding to at least one dataset for training. The paired datasets were collected from the same experiments. Thus, estimated parameters on priming from the training dataset can be evaluated with the validation dataset. In this sense, our previous validation used fixed parameters during the validation (or called it a fixed mode of validation). To broadly test whether the estimated priming can be validated by any independent experiments, we conducted another validation (or called it a random mode of validation) in the revised manuscript. The collected data sets were randomly divided into two groups, one for model training and the other for validation. The two groups had similar distribution of the added new C amount (% of SOC) (Supplementary Figure 3 and below, panel a is for the training and panel b is for the validation).

The trained interactive model by the first group of data was run to predict the priming effect of the second group of studies, and was compared with the observations. The results showed that the model predictions and observations have very similar distributions (Supplementary Figure 4 and below; the black stairs and the red bars are modeled and observed, respectively), indicating that estimated priming with the interactive model well represents decomposition of old SOC.

In the revised manuscript, we added sentences on lines 146-150 to describe the results as:

“The interactive model fitted data well, was the most parsimonious model (with the smallest DIC; Fig. 4b, e), and was further validated by two modes with either fixed or randomized parameters (i.e., statistically called out-of-sample validation; Supplementary Figs. 2-4). The validation with either fixed or randomized parameters indicated that the calibrated interactive model well represents the priming effects regardless of experimental conditions.”

We also added a new section under Methods to describe the details (lines 352-369) as

“Model validation. *To further validate the selected model (i.e., interactive model in this study) as an assurance of model extrapolation beyond the observations, we employed two modes with both fixed and random parameters for model validation. In the fixed mode of validation, we used three collected publications in which different amounts of new C were added into the same soils¹⁻³. For those studies, one new C amount was used for model selection and parameter optimization (studies 8, 34, 36, 42 in Supplementary Table 1; training group), and the others were used for model validation (studies 9, 10, 35, 37, 43 in Supplementary Table 1; validation group). The interactive model with optimized (i.e., fixed) parameters with the training data was run with the new C amount at the validation group, and the modeled decomposition rates of SOC and added substrates were compared with observed ones (Supplementary Fig. 2). In the random mode of model validation, the collected 84 datasets were randomly divided into two groups, one for model training and the other for validation. The two groups had similar distribution of the added new C amount (% of SOC) (Supplementary Fig. 3). The trained interactive model by the first group of data was run to predict the priming effect of the second group of data, and was compared with the observations (Supplementary Fig. 4). The model selection and validation results indicated that the selected interactive model had the ability to represent SOC decomposition with priming and replenishment. Thus, the interactive model was used for further analyses.”*

In the previous version, the collected experiments lasted for 28-168 days. In the literature, there is an experiment lasted for 209 days⁶, which may be the one the reviewer referred to. We did not include this experiment in the last version because it did not report the CO₂ flux from the added substrate. In the revised manuscript, we included the datasets from that experiment in our analysis, which did not change our conclusions. Thus, we included those datasets, and there are totally 84 datasets in the revised manuscript.

Lines 112-117. When evaluating the data set, it appears there is a highly significant linear relationship between net SOC change and the amount of C added. Thus, the replenishment effects are significantly influenced by the amount added. Was the amount added in these studies justified based on certain criteria? The authors do standardize the results by the amount of substrate added, but even within one soil type one may expect different responses based on the amount of substrate added. Could this be influencing the overall effect and conclusions made in this synthesis?

Response: This reviewer is very knowledgeable and asked a great question. To address this comment, we plotted the loss of added C, replenishment and net SOC change against the added C amount. Results showed that the amount of added substrate did not have significant effects on

the loss of the added C, replenishment, or net SOC change as presented below and in Supplementary Fig. 5.

To address reviewer’s question “*Was the amount added in these studies justified based on certain criteria?*”, we read those papers again to examine whether scientists who conducted those original experiments used any criteria to justify the amount of C added in their studies. We did not see any common criteria across those studies.

Lines 142-144. Unclear sentence. What is meant by "complex SOC"? It is unclear how the model inadequately described priming with complex SOC. Can complexity be ascribed to simple C:N ratios? What is meant by "mono-substrates"?

Response: We apologize for the confusion caused by those terms in the last version. To avoid the confusion, we removed the two terms, and revised the sentence on lines 139-143:

“The model evaluation against all the training data (group I in Supplementary Table 1; statistically called within-sample evaluation) indicated that the regular Michaelis-Menten model did not adequately describe the SOC decomposition with priming and replenishment, showing a relative high Deviance information criterion (DIC) and an extremely low likelihood of model given the data (Fig. 4c and e).”

Lines 150-152. The claim that this particular model can be used for ecosystem and earth-system models is too far reaching. It is unclear how the data based on laboratory incubations is being extrapolated to the ecosystem. How are different climatic variables and soil properties being used? Would introducing this complexity strengthen or weaken these results?

Response: We greatly appreciate the reviewer for pointing this issue out. We can understand reviewer’s concern on using the interactive model for ecosystem and earth-system models. As a consequence, we deleted the sentence and downplayed the point on potential uses of the interactive model for ecosystem and earth-system modeling in this revision. Our conclusion was limited to the degree as in the following sentence and on lines 211-213:

“Our validation of the model with either fixed or randomized parameters indicates that the interactive model is able to well represent the priming effect and replenishment regardless of experimental conditions.”

We agree with the reviewer that many factors, such as “*climatic variables and soil properties*”, need to consider when upscaling from the laboratory to the ecosystem scales. We made additional analysis to explore how temperature and moisture fluctuations influence net SOC change as shown below and Supplementary Figure 9.

We also examined how soil properties, such as soil moisture and temperature influence priming, replenishment, and net carbon change. Since we did not find any relationships among them as shown below and Supplementary Figure 6, we did not explore how soil properties influence upscaling. Anyhow, upscaling is not the main focus of this study.

In the manuscript, we also discussed the potential limitation of upscaling the interactive model for the laboratory to larger scales on lines 197-202:

“In this study, the quantitative estimations were based on laboratory incubation experiments, which may be biased when applying in the field due to at least the two following reasons. First, disturbance and micro-environmental changes in the incubation experiments may influence the magnitudes of the replenishment, priming and net effect. Second, soil microbial community in the incubation jars may be different from that in the field. Thus, the values of the replenishment, priming and net effect reported in this study should be used with caution.”

Lines 163-180. There are a lot of details missing from the description of the 100-year modeling experiment. First, I am not convinced that that the laboratory data is appropriately used to describe long-term dynamics. There are a number of studies that show how temperature and soil

moisture can significantly influence the magnitude of soil C loss. How was this varied in the 100-year modeling exercise? What about N?

Response: This reviewer's concern on whether or not the laboratory data can be used to describe long-term dynamics is very similar to comment 5 by reviewer 1. Indeed, this concern is very common in the global change research community.

If we carefully examine ecosystem or earth system models, many of their components are based on short-term experiments. For example, the leaf photosynthesis model by Farquhar et al. (1980)⁵ describes short-term, fast biochemical processes. Those processes were mostly studied in laboratory. However, Farquhar model, which synthesizes fast biochemical processes from short-term experiments, has been used at time scales from minutes to centuries and at spatial scales from biochemistry to the globe. If we believe those short-term experiments included in our study reveal fundamental processes underlying priming and replenishment, our synthesis of those experimental results into models shall be valid in term of representing those processes in the model.

Nevertheless, to be responsive to reviewer's concern, we have deleted the sentence on potential uses of the interactive model for ecosystem and earth-system modeling in this revision. Our conclusion was limited to the degree as in the following sentence and on lines 211-213:

“Our validation of the model with either fixed or randomized parameters indicates that the interactive model is able to well represent the priming effect and replenishment regardless of experimental conditions.”

Thanks to reviewer's comment, we removed the 100-year simulation in the revised manuscript. Instead, we conducted a short-term (i.e., 1 year) modeling incubation experiment to analyze the effect of continuously increased C addition on the SOC change (Fig. 6).

We also agree with the reviewer on the point that *“temperature and soil moisture can significantly influence the magnitude of soil C loss”*. The reviewer is correct on the point that there are many studies on temperature and moisture effects on soil C loss. In this study, we primarily focused on the effect of increased new C input on SOC content. Both the synthesis of data (Fig. 2) and the modeling experiment with continuous C input (Fig. 6) were to illustrate how increased new C input impact SOC content through priming and replenishment. Such modeling exercise was intended to illustrate impacts of priming vs. replenishment on soil C dynamics and has been commonly done before for illustration. To be responsive to reviewer's suggestion, we have conducted additional analyses on N influences and with fluctuating temperature and moisture.

As illustrated in Fig. 5 and Supplementary Fig. 8, N does influence the magnitude of priming and replenishment of SOC. The magnitude of net SOC change generally increases with the increase in substrate N:C ratio, under both one-time (Fig. 5) and continuous C input conditions (Supplementary Figs. 8 and below, panels **a** and **b** show the results with step and gradual increase in C inputs, respectively. The low, medium and high N:C ratios are corresponding to the groups in Fig 5).

Our analysis indicates that fluctuating temperature and moisture do not significantly influence modeled SOC sequestration (Supplementary Fig. 9 and below; Panels a and b are for step and gradual increases, respectively).

Lines 182-186. Why not explore these mechanisms further? Could the range of soil and climatic characteristics from this data set be used to explore the factors regulating SOC dynamics?

Response: We greatly appreciate the reviewer for the suggestions. We agree that soil and climatic characteristics may regulate SOC dynamics.

Per reviewer's question above, we made additional analyses and found that neither incubation temperature nor moisture had any clear influences on the SOC change after new C addition (Supplementary Fig. 6 and below).

Although many mechanisms, including physical and chemical bonding of new C to the soil mineral complex and SOC formation through microbial metabolic processes, have been proposed in the literature, the experiments mostly did not report any measurements related to physical/chemical bonding or microbial metabolic processes. However, citing those publications with proposed mechanisms enriches discussion in this manuscript and is generally in accordance with our conclusions.

Lines 196-200. Why not expand on this? If the studies also measured soil nutrients you should see this in the data set.

Response: In this study, we observed that a higher priming loss of old SOC occurred when the added substrates have lower N:C ratios (Fig. 5). This part (lines 186-196 in the revised manuscript) was to discuss that this was likely due to microbial nitrogen mining from old soil organic matter when the added substrate was nitrogen limited. The reviewer suggested that we may expand on this. We agree that expanding on this would strengthen the discussion. However, most experiments synthesized in this study did not report nitrogen mineralization data. Although we did not have direct data from the collected experiments, previous studies^{7,8} indicated that nitrogen mining for microbial growth is an important mechanism of the priming effect when the added substrate is nitrogen limited. Citing those publications with proposed mechanisms enriches discussion in this manuscript and is generally in accordance with our conclusions. This idea deserves exploration in the future study. To emphasize this point, we added one sentence on lines 191-193 as below:

“To further confirm the N mining hypothesis, we need more innovative incubation experiments to simultaneously quantify both C loss and N mineralization in response to additions of new C with different N content.”

Lines 213-216. This conclusion is a stretch and I am not convinced based on this study one can conclude what is best for ecosystem and Earth system models.

Response: The reviewer raised a similar concern above. To be responsive to reviewer’s concern, we have deleted the sentence on potential uses of the interactive model for ecosystem and earth-system modeling in this revision. We went with our conclusion as far as in the following sentence and on lines 211-213:

“Our validation of the model with either fixed or randomized parameters indicates that the interactive model is able to well represent the priming effect and replenishment regardless of experimental conditions.”

Lines 330-332. ‘100 sets of parameters’ - It is unclear what this means as written.

Response: In the revised manuscript, the long-term simulation was replaced by a short-term (1 year) modeling experiment. We have deleted this term to avoid confusion.

Supplementary Table 1: It would be informative to include the location from which these soils were sampled.

Response: We have added the location information, including latitude/longitude and country, as suggested (Supplementary Table 1).

Reviewer #3 (Remarks to the Author):

The paper is an interesting study of carbon priming affects on soil organic carbon stocks, a topic of great relevance to earth system models. The study combines a meta-analysis of empirical studies from a broad range of systems in relation to four types of models that are commonly used to represent SOC dynamics. The study is significant for at least two reasons. First, the replenishment to priming ratios show a relatively narrow range despite the variation in soil types and added substrate composition with direct application. Second, the exercise shows that the most parsimonious model (interactive) offers the greatest predictive value. There are frequent calls in the literature for inclusion of more microbial detail in earth system models. This study illustrates that such detail, which may be difficult acquire, may not be helpful. The study is thorough and well described. It is difficult to digest given the need to keep referring to supplemental materials and the epilogue methods section for relevant context.

Response: We are pleased that the reviewer found our study significant and interesting.

Line 159-165. The N:C result is consistent with the N mining hypothesis that labile carbon inputs stimulate the degradation of SOC for N acquisition. This topic is raised again in the discussion (line 199). This interaction of C priming and N mining is interesting. The authors might consider expanding on this topic. With the data on hand, it appears possible to estimate the potential N mineralization from SOC in relation to substrate N input. This exercise is analogous to the C input – C mineralization results presented, and might provide additional insight into the controls of SOM accumulation.

Response: We agree that expanding on this topic would strengthen the discussion. Unfortunately, most experiments synthesized in this study did not report nitrogen mineralization. Without measurements, we could not make any assessment on how priming is linked with nitrogen mineralization. Although we did not have direct data from the collected experiments, previous studies^{7,8} indicated that nitrogen mining for microbial growth is an important mechanism of the priming effect when the added substrate is nitrogen limited. Citing those publications with

proposed mechanisms enriches discussion in this manuscript and is generally in accordance with our conclusions.

This reviewer's suggestion deserves exploration in the future study. To emphasize this point, we added one sentence on lines 191-193 as below:

“To further confirm the N mining hypothesis, we need more innovative incubation experiments to simultaneously quantify both C loss and N mineralization in response to additions of new C with different N content.”

References:

- 1 Blagodatskaya, E., Yuyukina, T., Blagodatsky, S. & Kuzyakov, Y. Three-source-partitioning of microbial biomass and of CO₂ efflux from soil to evaluate mechanisms of priming effects. *Soil Biol Biochem* **43**, 778-786, doi:DOI 10.1016/j.soilbio.2010.12.011 (2011).
- 2 Guenet, B., Leloup, J., Raynaud, X., Bardoux, G. & Abbadie, L. Negative priming effect on mineralization in a soil free of vegetation for 80 years. *Eur J Soil Sci* **61**, 384-391 (2010).
- 3 Wu, J., Brookes, P. C. & Jenkinson, D. S. Formation and Destruction of Microbial Biomass during the Decomposition of Glucose and Ryegrass in Soil. *Soil Biol Biochem* **25**, 1435-1441 (1993).
- 4 Spiegelhalter, D. J., Best, N. G., Carlin, B. P. & Van Der Linde, A. Bayesian measures of model complexity and fit. *Journal of the Royal Statistical Society: Series B* **64**, 583-639 (2002).
- 5 Farquhar, G. v., von Caemmerer, S. v. & Berry, J. A biochemical model of photosynthetic CO₂ assimilation in leaves of C₃ species. *planta* **149**, 78-90 (1980).
- 6 Guenet, B., Juarez, S., Bardoux, G., Abbadie, L. & Chenu, C. Evidence that stable C is as vulnerable to priming effect as is more labile C in soil. *Soil Biol Biochem* **52**, 43-48 (2012).
- 7 Chen, R. *et al.* Soil C and N availability determine the priming effect: microbial N mining and stoichiometric decomposition theories. *Global Change Biol* **20**, 2356-2367 (2014).
- 8 Kuzyakov, Y., Friedel, J. K. & Stahr, K. Review of mechanisms and quantification of priming effects. *Soil Biol Biochem* **32**, 1485-1498 (2000).

REVIEWERS' COMMENTS:

Reviewer #2 (Remarks to the Author):

The authors adequately addressed my previous comments. I believe the paper has improved with the added detail, further clarifications, removal of the long-term (100 year) model simulation, and the removal of the over-reaching claims. I offer some comments below where further clarification or textual edits are needed.

In response to my previous comment about how realistic the substrate additions were, which ranged from <1% to 35% of SOC stocks, you stated that litter inputs are 3-5% of global SOC stocks. This is at the very low end of the range of substrate additions used. Also, much of the single substrate additions have no relation to litter inputs. I think it would be good to include text that explicitly relates the experimental additions and current or projected C input estimates via enhanced productivity—something to put your results into context.

Line 142: "A low likelihood of model"... meaning a low likelihood of data-model agreement or data representation, right? I would add text to clarify.

Lines 167: Specify where the 10% step increase starting from.

Line 171-172: "SOC storage" insinuates long-turnover time, of which this study does not address. Instead, it would be more appropriate to state "accumulation of added substrate". Also, instead of stating, "even with fluctuating temperature and moisture" it would be more appropriate to say "independent of temperature and moisture conditions". To me, "fluctuating" insinuates that temperature and moisture were manipulated in each experiment.

Lines 174: As stated previously, the text "SOC storage" is not necessarily appropriate. This study depicts the amount of new substrate lost via microbial respiration (CO₂) versus that which accumulates in the soil, but does not explicitly measure or model the stabilization potential. In other words, there is no evidence to suggest that this new substrate is stabilized and will be stored for longer-term. I would change the language throughout so that it is more directly related to your results, which should not be interpreted as long-term storage

Reviewer #4 (Remarks to the Author):

This is an interesting and relevant synthesis of incubation priming experiments. The modeling extrapolates the effects of priming and replenishment over a reasonable time frame and the relationship observed between N:C is well supported ecological theory. The authors do a thorough job of addressing the reviewer's concerns, particularly regarding validation and correlations with variables such as amount of new C input.

I have a few comments/questions that I hope are helpful:

1) Three of the models used describe the transfer of new C to old C. I'm not totally sure what old C is. In the context of a priming experiment, I assume it is the pre-existing SOM that can be decomposed, but in the context of a model that transfers new C into this pool, it then seems more like "stable" SOM or "microbially-processed" SOM. A little clarification here would be helpful.

2) Are the differences in SOC change observed between the N:C ratio categories significant? I think it is an interesting trend either way, but if there was a statistical test I missed it.

3) Because the models do not include soil moisture effects directly, the Supp Fig 9 analysis is more

of a parametric sensitivity analysis when it comes to soil moisture effects. I don't think your conclusions rely on this being a good test of soil moisture effects, except at L172, where you could perhaps remove the clause, "even with fluctuating temperature and soil moisture".

Note: I wrote some theoretical modeling papers as part of my PhD where priming effects were dependent on N:C of the substrate. We didn't have much data to support this theory, so I am glad to see it here!

Best,
Rose Abramoff

Letter of Responses

Authors' Note: The original reviewers' comments are *in italic and colored blue*, and our responses follow. All line numbers indicated in the responses are those in the revised manuscript.

REVIEWERS' COMMENTS:

Reviewer #2 (Remarks to the Author):

The authors adequately addressed my previous comments. I believe the paper has improved with the added detail, further clarifications, removal of the long-term (100 year) model simulation, and the removal of the over-reaching claims. I offer some comments below where further clarification or textual edits are needed.

Response: We greatly appreciate the constructive comments by the reviewer during the peer review process.

In response to my previous comment about how realistic the substrate additions were, which ranged from <1% to 35% of SOC stocks, you stated that litter inputs are 3-5% of global SOC stocks. This is at the very low end of the range of substrate additions used. Also, much of the single substrate additions have no relation to litter inputs. I think it would be good to include text that explicitly relates the experimental additions and current or projected C input estimates via enhanced productivity—something to put your results into context.

Response: The reviewer provided a great suggestion. The reviewer concerned that the global litter productivity “*is at the very low end of the range of substrate additions used*”. As shown in Supplementary Figure 3, although the amount of substrate additions ranged from <1% to 35% of SOC stocks, most (i.e., over 2/3) of them fell within the range of < 10% of SOC stocks. Global litter productivity is about 3-5% of global SOC stocks. The total C input to soils would be even more if considering root exudates though the global estimate is uncertain to our knowledge. In addition, Earth systems models generally predict the C input could increase by 25% to 60% by the end of the 21st century (Zhou et al., 2018). Thus, the experimental additions are generally in accordance with the global C input estimates.

In the revision, we added text to relate the experimental additions and carbon input estimates (lines 305 – 311):

“The amount of added C in most (i.e., over 2/3) of the collected studies fell within the range of < 10% of SOC stocks. Global litter productivity is about 3-5% of global SOC stocks. The total C input to soils would be even more if considering root exudates though the global estimate is uncertain to our knowledge. In addition, Earth systems models generally predict the C input could increase by 25% to 60% by the end of the 21st century⁵³. Thus, the experimental additions are generally in accordance with the global C input estimates.”

In addition, we standardized all the results by the amount of added C, which partly eliminated the effect of the substrate amount on the results.

Line 142: “A low likelihood of model”... meaning a low likelihood of data-model agreement or data representation, right? I would add text to clarify.

Response: Yes. By saying “a low likelihood of model”, we meant a low data-model agreement. In the revision, we revised the sentence (lines 189 – 193) as “*The model evaluation against all the training data (group I in Supplementary Data 1; statistically called within-sample evaluation) indicated that the regular Michaelis-Menten model did not adequately describe the SOC decomposition with priming and replenishment, showing a relative high Deviance information criterion (DIC) and a low data-model agreement (Fig. 4c; Table 1).*”

Lines 167: Specify where the 10% step increase starting from.

Response: Revised as suggested. The 10% step increase started from the beginning of the modeling experiment. The revised sentence (lines 229 – 231) is “*Results showed that a 10% step increase in C input starting from the beginning of the modeling experiment enhanced SOC by 43.1% of the total increased C input after one year (Fig. 6a).*”

Line 171-172: “SOC storage” insinuates long-turnover time, of which this study does not address. Instead, it would be more appropriate to state “accumulation of added substrate”. Also, instead of stating, “even with fluctuating temperature and moisture” it would be more appropriate to say “independent of temperature and moisture conditions”. To me, “fluctuating” insinuates that temperature and moisture were manipulated in each experiment.

Response: Revised as suggested. The sentence (lines 236 – 238) is revised to “*Overall, the modeling experiment confirmed that increased new C inputs promote accumulation of added substrates, which was independent of temperature and moisture conditions (Supplementary Fig. 9).*”

Lines 174: As stated previously, the text “SOC storage” is not necessarily appropriate. This study depicts the amount of new substrate lost via microbial respiration (CO₂) versus that which accumulates in the soil, but does not explicitly measure or model the stabilization potential. In other words, there is no evidence to suggest that this new substrate is stabilized and will be stored for longer-term. I would change the language throughout so that it is more directly related to your results, which should not be interpreted as long-term storage

Response: Revised as suggested. The revised sentence (lines 241 – 242) is “*The general C accumulation after the additional new C input may be due to both physiochemical and biological interactions.*”

In addition, we revised the language throughout the manuscript as suggested.

Lines 76 – 78: “Our findings suggest that increasing C input to soils likely promote SOC accumulation despite the enhanced decomposition of old C via priming.”

Lines 249 – 251: “Despite the general pattern of C accumulations following a new C input (Fig. 2), several individual studies have shown net SOC loss, primarily from saline alkaline²¹ or low-fertility soils²².”

Lines 276 – 277: “Overall, our study evaluated the effect of new C addition on SOC accumulation through two critical processes: priming and replenishment.”

Reviewer #4 (Remarks to the Author):

This is an interesting and relevant synthesis of incubation priming experiments. The modeling extrapolates the effects of priming and replenishment over a reasonable time frame and the relationship observed between N:C is well supported ecological theory. The authors do a thorough job of addressing the reviewer’s concerns, particularly regarding validation and correlations with variables such as amount of new C input.

Response: We appreciate the positive comments.

I have a few comments/questions that I hope are helpful:

1) Three of the models used describe the transfer of new C to old C. I’m not totally sure what old C is. In the context of a priming experiment, I assume it is the pre-existing SOM that can be decomposed, but in the context of a model that transfers new C into this pool, it then seems more like “stable” SOM or “microbially-processed” SOM. A little clarification here would be helpful.

Response: The reviewer provided an insightful comment. In the models, old C pools were those pre-existing and relative stable SOC, and new C pools were freshly added C which can be transferred to old C pools as decomposition proceeded. In the revision, we added a sentence to clarify (lines 327 – 329):

“In the models, old C pools were those pre-existing and relative stable SOC, and new C pools were freshly added C which can be transferred to old C pools as decomposition proceeded.”

2) Are the differences in SOC change observed between the N:C ratio categories significant? I think it is an interesting trend either way, but if there was a statistical test I missed it.

Response: Yes, the differences in SOC change between the N:C ratio categories were significant because the 95% confidence intervals were not overlapped (Fig. 5). In the revision, we revised the figure caption to add more information (Lines 771 – 775):

“Figure 5 / Synthesis of the dependence of annual replenishment, priming and net SOC change on substrate N:C ratio. The replenishment increased, but priming decreased, with the increase in substrate N:C ratio. Thus, the net SOC change significantly increased with the increase in substrate N:C ratio. The number of studies for each category is shown near the bar. Mean ± 95% confidence interval.”

3) Because the models do not include soil moisture effects directly, the Supp Fig 9 analysis is more of a parametric sensitivity analysis when it comes to soil moisture effects. I don't think your conclusions rely on this being a good test of soil moisture effects, except at L172, where you could perhaps remove the clause, "even with fluctuating temperature and soil moisture".

Response: We appreciate the detailed comment. By considering the comments from both reviewers, we revised the sentence (lines 236 – 238) to “*Overall, the modeling experiment confirmed that increased new C inputs promote accumulation of added substrates, which was independent of temperature and moisture conditions (Supplementary Fig. 9).*”

Note: I wrote some theoretical modeling papers as part of my PhD where priming effects were dependent on N:C of the substrate. We didn't have much data to support this theory, so I am glad to see it here!

Response: We are glad this synthesis supports the theory that priming effects are dependent on N:C ratio of the substrate.